# The structure, centrality, and scale of urban street networks: Cases from Pre-Industrial Afro-Eurasia

Mark Altaweel[1]*, Jack Hanson[2], Andrea Squitieri[3]

**1** Institute of Archaeology, University College London, London, United Kingdom, **2** Department of Classics, University of Reading, Reading, United Kingdom, **3** Institut für Ur- und Frühgeschichte und Vorderasiatische Archäologie, Universität Heidelberg, Munich, Germany

* m.altaweel@ucl.ac.uk

## Abstract

Cities and towns have often developed infrastructure that enabled a variety of socio-economic interactions. Street networks within these urban settings provide key access to resources, neighborhoods, and cultural facilities. Studies on settlement scaling have also demonstrated that a variety of urban infrastructure and resources indicate clear population scaling relationships in both modern and ancient settings. This article presents an approach that investigates past street network centrality and its relationship to population scaling in urban contexts. Centrality results are compared statistically among different urban settings, which are categorized as orthogonal (i.e., planned) or self-organizing (i.e., organic) urban settings, with places having both characteristics classified as hybrid. Results demonstrate that street nodes have a power law relationship to urban area, where the number of nodes increases and node density decreases in a sub-linear manner for larger sites. Most median centrality values decrease in a negative sub-linear manner as sites are larger, with organic and hybrid urban sites' centrality being generally less and diminishing more rapidly than orthogonal settings. Diminishing centrality shows comparability to modern urban systems, where larger urban districts may restrict overall interaction due to increasing transport costs over wider areas. Centrality results indicate that scaling results have multiples of approximately ⅙ or ⅓ that are comparable to other urban and road infrastructure, suggesting a potential relationship between different infrastructure features and population in urban centers. The results have implications for archaeological settlements where urban street plans are incomplete or undetermined, as it allows forecasts to be made on past urban sites' street network centrality. Additionally, a tool to enable analysis of street networks and centrality is provided as part of the contribution.

## Introduction

Measures for street network centrality have been used to understand wider social interactions. The application of centrality measures includes modern and ancient urban contexts, where

**Data Availability Statement:** Data and code for this work can be downloaded here: https://doi.org/10.5522/04/15191601.

**Funding:** The Center of Advanced Studies-Schwerpunkt (Siedlungen zwischen Diversität und

Homogenität) from the University of Munich (LMU) provided funding for this effort.

**Competing interests:** The authors have declared that no competing interests exist.

researchers attempt to understand structural and functional aspects of urban street networks [1–4]. Such analyses provide insights into how settlements enable social interaction and accessibility of goods and services, such as ease of access to markets, work sites, religious venues, or even entertainment. Scholars have also deployed measures of urban scaling to understand how populations shape and are affected by the growth of urban infrastructure. Settlement scaling approaches have been used in archaeology as a means to better understand how a variety of social interactions, including related to trade and information flow, are shared within urban spaces [5–8]. Theory on settlement scaling suggests that urban infrastructure, in the past or present, should demonstrate sub-linear growth to population [5, 8–10]. This suggests that street networks should demonstrate population scaling properties comparable to other infrastructure features. If this is the case, these properties should enable one to reasonably estimate street centrality values for given urban contexts, even in cases where only part or minimal areas are known.

This article proposes to evaluate urban street structures for pre-industrial urban sites in Europe, North Africa, and the Middle East, spanning periods from the Middle Bronze Age (c. 2000–1800 BCE) to the early Modern Period (i.e., around the 18[th] century CE). It analyzes urban street layouts and their centrality values, using a tool developed by this article and made available in the supporting materials, measuring how they scale relative to urban area estimates, that is occupation size for a given period. In this case, area is used as a proxy for population. We determine the degree to which power law relationships compare to street network centrality. Urban locations are assessed based on their street organization, including orthogonal (i.e., rectilinear, planned or grid-pattern streets), organic, that is self-organizing streets that generally develop around neighborhoods, and hybrid streets, which have a combination or mixture of orthogonal and organic street networks [11]. The intent is to indicate if settlements in pre-industrial societies displayed common accessibility and centrality properties for different types of urban settings found in regions as well as different periods. A benefit of the approach is that results from this work could be utilized to build a model for estimating centrality for archaeological urban settings where street networks are missing or incomplete. The approach further provides insights and estimates into relative accessibility and communication links within urban settings, with centrality providing insight into the degree that given urban settings facilitated social interaction. In order to evaluate centrality and determine its population scaling properties, with results used to reinforce each other, multiple centrality measures are applied to understand street networks, including: betweenness, closeness, degree, efficiency, eigenvector, harmonic, Katz, straightness, and current flow. These centrality measures are compared for the different types of urban settings analyzed, with orthogonal and organic or hybrid used as the two main analytical categories due to structural similarities between hybrid and organic sites.

To begin this study, background information on street network analysis, including space syntax methods, and urban scaling are presented. The data used for this study are then given. The next section details the methods incorporated, including the scaling and network centrality methods applied here. Results for network centrality distributions, centrality, and urban area scaling are provided based on different urban street layout types. This includes the evaluation of orthogonal, or grid-planned street networks, organic, or neighborhood organized or non-centrally organized streets, and streets that are a combination of these types (i.e., hybrid). In the discussion and conclusion section, the outputs and their implications for urban theory are developed. This includes how such theory can be used to understand different urban settings in varied ancient and more recent periods, including where urban street layouts are not well known. Future research and limitations of this work are also suggested.

## Background

### Space syntax analysis and street networks

Street network centrality measures are part of wider space syntax analysis. The application of space syntax analysis has been used for decades, since at least the 1970s, to better comprehend architecture and urban layouts [12]. In archaeology, space syntax has been used to understand past urban spatial partitioning, dimensions of urban spaces, and how urban spaces affect sensory perceptions [1, 13–15]. Even before formal space syntax analysis, Giddens [16] demonstrated how space is, along with time, among the main contributors to an agent's social interactions. Craane [17] has summarized the characteristics of space syntax. It is described as a set of techniques used for analyzing urban spaces as networks, looking at the placement, grouping, and orientation of buildings. Patterns of how networks of space are used are also analyzed, such as land use, transport, or security. Linking it to theory, space syntax reflects ideas on urban space networks' reflection and relation to social, economic, and cognitive factors shaped by space.

In the analysis of street networks, a variety of techniques to understand urban and non-urban roads are used, with graph analysis being the most common set of methods applied for understanding street network relationships and relationships of regions and sub-regions in defined settings [18, 19]. Common techniques employ node centrality measures, drawing from sociology or social network analyses, to study the relevance of spaces or traffic patterns for social activity [20–23]. In archaeology, formal studies investigating centrality for street networks are relatively rare, particularly because in many periods sites are not adequately preserved to allow clear reconstruction of ancient streets. Many approaches, such as work by Bikoulis [24] and Brughmans [25], have utilized network centrality methods on regional site interactions rather than on site-specific transport contexts. However, a few studies at the site level exist. Poehler [26] investigates movement in Pompeii using betweenness centrality to find economically attractive spaces that could benefit from the city's urban layout. In the ancient settlement of Kerkenes Dağ in Anatolia, Altaweel and Wu [27] applied an agent-based model and Branting [28] applied a GIS-Transportation (GIS-T) to investigate likely streets with the highest ancient traffic. These outputs and approaches were validated through geoarchaeological fieldwork that demonstrated forecasted streets with relatively higher ancient traffic volume in the past. These methods effectively replicate betweenness centrality in understanding ancient traffic, where it was likely to have concentrated, while determining likely spaces for public or private social interaction. Overall, few past urban street layouts are known to a high enough level that facilitates any analytical approach that can determine key spatial relationships and understanding of street networks, limiting our understanding of interactions and traffic patterns within most urban settings.

### Urban scaling

While the knowledge that populations and resources show linear, sub-linear, and super-linear scaling relationships has been well known for some time for modern urban contexts [10], it has become clear that past urban systems have comparable scaling results for related urban phenomena [5, 29]. Archaeological investigations focusing on population scaling relationships to urban phenomena have researched such topics as included: residential unit densities [30], the dimensions of mixing spaces, such as public spaces and street networks [9], city gate sizes [31], urban structures [32], inter-urban transport networks [33], social connectivity and material flows [34], economic returns [29], and labor activities [6]. Work has also demonstrated that population relationships appear to indicate scaling relationships to measured urban areas,

where results have supported that throughout history socioeconomic networks have structured relationships to urban spaces [7, 30]. Although some work has been done on the relationships between street networks and estimated populations for sites, previous work has only considered the lengths and widths of streets, without considering their internal network structure or performing a formal analysis of internal networks [9]. In other words, one topic missing from current discussions is whether there is any relationship between the centrality of past street networks and urban areas. From referenced works above and other research, the examples show that scaling properties are likely evident for a wide range of urban phenomena, including urban street properties and urban areas [35]. Research on modern street networks has demonstrated not only scaling relationships, but those relationships are affected by how places are shaped, including the geometry and street layout of cities [36]. For modern cities, work has shown that many urban street networks show comparable scaling relationships for centrality measures in different types of cities. However, for certain centrality metrics, including information centrality, planned cities (i.e., orthogonal-shaped cities) show exponential relationships. On the other hand, a power-law scaling relationship is observed for organic or self-organized urban spaces when using information centrality. Organic and planned cities generally have indicated some different centrality distributions [3]. Although modern cities have indicated some scaling relationships between population and street networks, it is an open question if past urban streets have comparable relationships and what the distributions of centrality might be.

## Materials and methods

### Definition of settlement types

First, we define the three types of settlements we use here. The first is orthogonal, which is defined as settlements having rectilinear streets that form grid-like patterns across the urban landscape [11, 37, 38]. Organic settlements are seen as those that derive from a bottom-up development. They appear to develop around neighborhoods, hence their growth lacks an overall direction that can be discerned. This concept, or rather analogy, of cities as organisms, which has been borrowed from biology, essentially suggests that they had varied systems that develop so as to interrelate. In such systems, urban streets develop around neighborhoods as the basic development area, which creates a more complex pattern of streets that could change direction or abruptly end. Hybrid settlements are those that have a combination of centralized, orthogonal streets and more organic appearing streets [11, 39]. Such settlements can be typical in long-lived towns, such as modern cities that have been occupied since the Medieval period or longer. There is no clear definition of what threshold classifies a settlement as hybrid but generally it is understood that these settlements have some combination of orthogonal and organic appearance.

### Case studies

For many regions, few complete plans exist for archaeological sites that allow street systems to be reconstructed. However, combining different regions and investigating over a wide time-scale aids to not only capture more complete street layout data but it also tests the idea that street systems scale to population in a manner comparable for different periods and regions. In fact, what is common to past, pre-industrial urban regions is that transportation options have been limited to animal or human-powered choices. In a variety of archaeological efforts, urban relationships have been compared by archaeologists by looking at a variety of regions and periods together, seeking common or comparable patterns in these cases to better understand urban phenomena and relationships [40–42]. We take a comparable approach here,

combining regions and periods to create a larger dataset than would otherwise be possible. Given this, street plans are collected here from the Middle Bronze Age (c. 1800 BCE) to the early Modern Period in the 18th century CE. The regions covered include North Africa, Europe, and the Middle East. Overall, 89 urban locations are collected, with data recorded to indicate the completeness of street networks. We classify site data based on if they are complete, that is if sites were 100% recoverable, mostly complete, where over 90% of the data are recoverable, and partial, with sites having less than 90% of their street networks likely evident. An example of orthogonal and organic streets is discussed by Yoo and Lee [43]. Orthogonal cities are defined as having rectilinear streets that form grid-like patterns across the urban landscape. For our purposes, sites are classified as having orthogonal or organic street layouts, using the definitions discussed above, based on if they show at least 20% of their area displaying these categories, with a hybrid layout reflecting that both these type classifications are evident in major urban sections. Data were collected based on how easy it was to find information, the expertise of the authors, and the availability of published street networks. The S1 File, with the data link provided, makes available sites used in this work and gives data on their area and street networks. The data list analytical summary outputs achieved, used in the following section, as well as references for street data. Additionally, the raw data and individual urban centrality outputs are provided. Fig 1 indicates urban sites and their locations studied here. No permits were required for the described study, which complied with all relevant regulations.

## Scaling approach

We test if there is a systematic relationship between median centrality measures, recovered from street networks where data can be sufficiently reconstructed, and a proxy for population, specifically area measured in hectares. Conceptually, our approach is comparable to other urban infrastructure and population research for modern and ancient settings that demonstrate systematic relationships between infrastructure, resources, and population or population proxies [6, 10, 29, 31, 33, 34, 44, 45]. The results help to offer a way to better estimate centrality in urban systems in the past, particularly as most ancient sites are only partially explored or unexplored, while outputs demonstrate insights into urban access and communication. In comparing street network centrality and area, we apply a scaling approach using what was applied in Lobo et al. [46]. This can be summarized as:

$$Y(N) = Y_0 N^\beta \tag{1}$$

Where $Y$ is median centrality expressed as having a scaling relationship to $N$, the size of the system or measured settlement area, and $\beta$ representing the scaling exponent, with $Y_0$ serving as a constant. This can be adjusted to accommodate statistical variation for sites using a log-transformed function expressed as:

$$lnY_i = lnY_0 + \beta lnN_i + \xi_i \tag{2}$$

Where $i$ is each indexed site area and $\xi_i$ reflects log deviations for sites in estimating median centrality $(Y)$. The value of this exponent, $\beta$, tends to be about ⅔ or ⅚, depending on context, at least in the case of measures of infrastructure (this means that these relationships deviate from linear by about ⅓ or ⅙). This variation has been explained by the different extents to which the built environments of settlements impose constraints on movement [8]. As a result, although there is currently no formal expectation for the relationship between the centrality measures discussed below and the sizes of sites, one might expect the slopes, or exponents, of these relationships deviate from linear by about ⅓ or ⅙ and that organic and planned

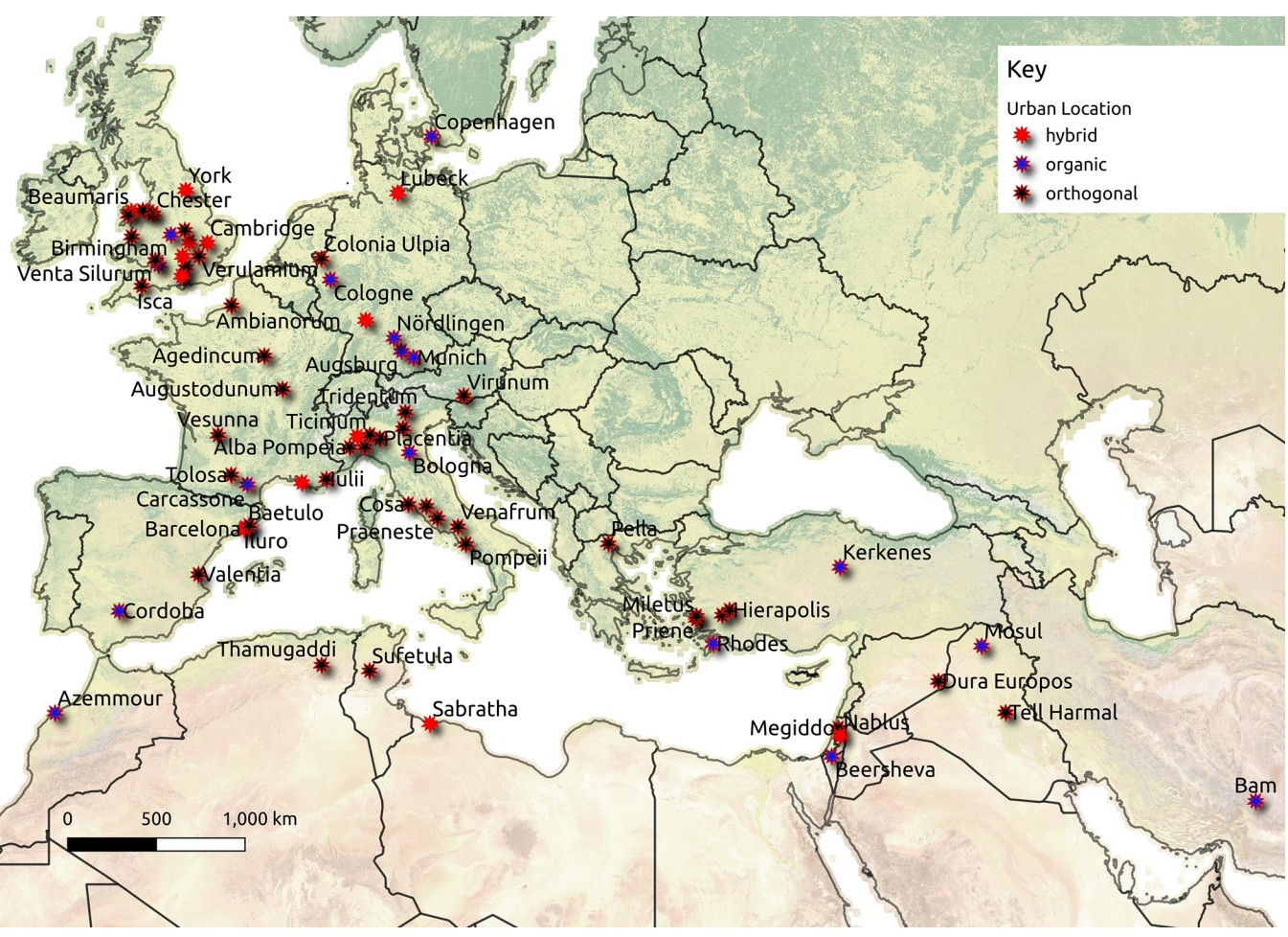

**Fig 1. Urban sites collected for this article.** Background map data courtesy of Natural Earth (https://www.naturalearthdata.com/).

settlements will take on different values. This is because other urban infrastructure and related phenomena, such as city gates and regional road networks [31–33], have demonstrated such values.

Before proceeding, it is important to make two points. The first is that, although most of the recent work on settlement scaling theory has used estimates for populations of sites, it is also clear that the inhabited areas of sites are related to both their densities and populations. This means that it is legitimate to explore the relationship between the attributes of sites and their inhabited areas, although the slopes of exponents for these relationships will be shallower than from those in other contexts. This is important in this study, given that it is not always possible to come up with reliable population estimates for sites. The second is that, although we would expect the baseline value of these relationships, which is represented by the y-intercept, or pre-factor, of these relationships, to vary from context, given that it reflects the prevailing social and economic conditions of the context in question, we would not expect this value to have changed significantly across the contexts that we are concerned with here. This is partly because we have concentrated on abstract measures of the built environment and partly because of the mutual dependence of our case-studies on the same or similar transportation technologies. This means that, although we would expect the geometry of the street network to be related to (i.e., a product of and a constraint on)city area, there is no reason for believing

that any of the network values that are discussed below should take on a specific baseline value or that this baseline value will change significantly over time. This suggests, in turn, that it is also legitimate to collapse examples from different historical and geographical periods into a single scaling relationship.

## Network analysis

While the above discussion reflects the scaling method deployed, networks also need to be constructed and assessed. In our approach, we utilized maps, satellite imagery, geophysical data, and published archaeological works with reconstructed urban street systems. In some cases, maps are used, which may not be accurate for measuring street links; however, maps are utilized for Medieval or early Modern towns. In these cases, these settlements also exist today and most of their streets are comparable to street plans present today. Therefore, imagery data are used to verify the location and distance of street links. Rather than true distances, relative distances within sites are maintained for analysis. Street networks are defined by nodes formed by street intersections [22, 23, 47]. Where streets turn substantially, that is over 30 degree turns, then a node was also created. In these cases, this represents a relatively sharp turn one would have to take that can justify a node or intersection, even if the street continues. Once street plans are recovered, they are mapped and links are assumed to be bi-directional. Since it is not possible to easily determine if past streets maintained a directional or bi-directional configuration, the intent is to measure centrality potential for urban plans rather than reconstruct centrality using likely traffic flows.

After street networks are reconstructed using GIS (see S1 File for data), network analysis is applied using NetworkX [48], a Python library deployed, and used within a Python analysis tool created by this effort, which is also provided in the S1 File, that also created additional network analysis. The total network analysis tool created (StreetCentrality) also outputs street network data into shapefile and.csv files used for visual and statistical analysis. Nine centrality measures are used in this work, which are: betweenness, closeness, degree, efficiency, eigenvector, harmonic, Katz, straightness, and current flow. Efficiency and straightness centrality are created within the StreetCentrality tool, while the others used are provided within NetworkX. The intent is to apply different centrality measures, that use distance and/or node connectivity as a measure, in order to see if distances between nodes or node connectivity show scaling relationships to settlement area. This potentially demonstrates how social and institutional access, that is streets that enable such access, scale within urban communities, where the results provide a variety of street network outputs that can be compared to demonstrate clear trends.

The following discussion reflects a summary of the node centrality methods applied with references given for the details on the algorithms applied. Betweenness centrality measures the number of shortest paths that pass through a vertex, where distance is used to calculate the shortest path. A node is more central if it has many shortest paths going through it. The applied algorithm follows Brandes' [49] implementation, with the results normalized to the number of nodes. Both nodes and edges are calculated for centrality, although nodes are used for scaling comparisons below. Closeness centrality measures the reciprocal for the distance of average shortest paths to all reachable nodes from a given node, where the results are also normalized [50, 51]. A variation of closeness centrality is harmonic centrality, which sums the reciprocal of the shortest path distances between nodes [52]. In harmonic centrality, nodes are more central if they are close to other nodes; however, a key difference is that harmonic centrality is not normalized for the number of nodes. Degree centrality represents the fraction of nodes a given node is connected to, with results normalized; this is based on how many links connect to a node [49]. Efficiency centrality, applied here, measures the ratio of the summed

multiplicative inverse of the shortest paths with the summed multiplicative inverse of the Euclidean distance for all nodes from given nodes. Straightness centrality is a variation of this, where the ratio of the Euclidean distance over the shortest path is summed for nodes and normalized [4, 53]. Both these measures reflect how much deviation there is between Euclidean and shortest path distances between nodes, reflecting how easy it is to move or transmit information in a network between nodes without a lot of extra movement getting to the desired destination. Eigenvector centrality calculates centrality for a node that is based on the connective centrality of neighboring nodes. In other words, a node becomes more central if other central nodes are found near it [54]. A more generalized variation of eigenvector centrality is Katz centrality, here normalized, where it measures centrality based on neighboring global and local centrality [55]. Both eigenvector and Katz imply that greater movement flows through places that are connected to other highly connected nodes, although Katz captures local and global influence and is normalized. Current flow centrality, or more specifically current flow closeness centrality, is equivalent to information centrality, where this measures centrality that is weighted by the inverse of path lengths. The harmonic mean lengths of paths that end at a vertex are smaller if the vertex has relatively short paths connecting to other vertices [56]. The measure is a type of hybrid that factors connectivity and distance-based measures in determining centrality since multiple paths are utilized in the measure. For all centrality measures, greater values indicate greater centrality.

## Results

### Centrality distributions

To demonstrate the street centrality approaches discussed above, Fig 2 presents measures for one site, Dura Europos [57], which originally dates to the Parthian-Roman period and was destroyed in the 3rd century CE. The figure uses mean values between nodes for edge centrality values as well as node centrality values for display. In this case, many centrality measures (Fig 2A, 2B, 2D–2F) demonstrate higher values near the *agora* (G1-G7), or main square, some of the key temples (H2, H4), or palace/temple area (C4/9). For some other centrality measures, results are more spread across the urban area. While the urban regions in which centrality values are relatively greater demonstrate some potentially relevant spaces for greater social interaction in the urban setting, this is not the focus here, since we are primarily interested in the patterns across sites. Rather, the results demonstrate the variations and distributions of relatively greater centrality based on distance and/or degree connectivity. As demonstrated here, degree-based measures (Fig 2C and 2G) indicate that orthogonal cities have somewhat even or minimal centrality variation, with some distance-based measure also demonstrating this (Fig 2H and 2I). The results show that there is generally greater centrality over a wide area in urban regions that are more reachable from varied areas, with most results also agreeing the central districts, in this case consisting of temples and the main *agora*, as being among the most central. Variations in centrality across the urban space are not always great, including when degree-based methods are utilized.

Fig 3 looks at late Medieval and early Modern Barcelona [58], a city that has prominent organic or self-organizing areas as well as orthogonal sections. While the old city region (Fig 3B, 3D and 3F), particularly near the cathedral, is prominent in centrality, other districts are also central in other measures, including areas with wider streets (Fig 3A, 3E and 3I). Some measures show similarity in centrality in different districts (Fig 3C, 3G and 3H). In contrast to Dura Europos, there is a more skewed concentration in centrality values when analyzing centrality differences (Fig 3A, 3E and 3I). While some degree-based measures show more evenness (Fig 3C and 3G) across the urban area than distance-based measures, similar to Dura

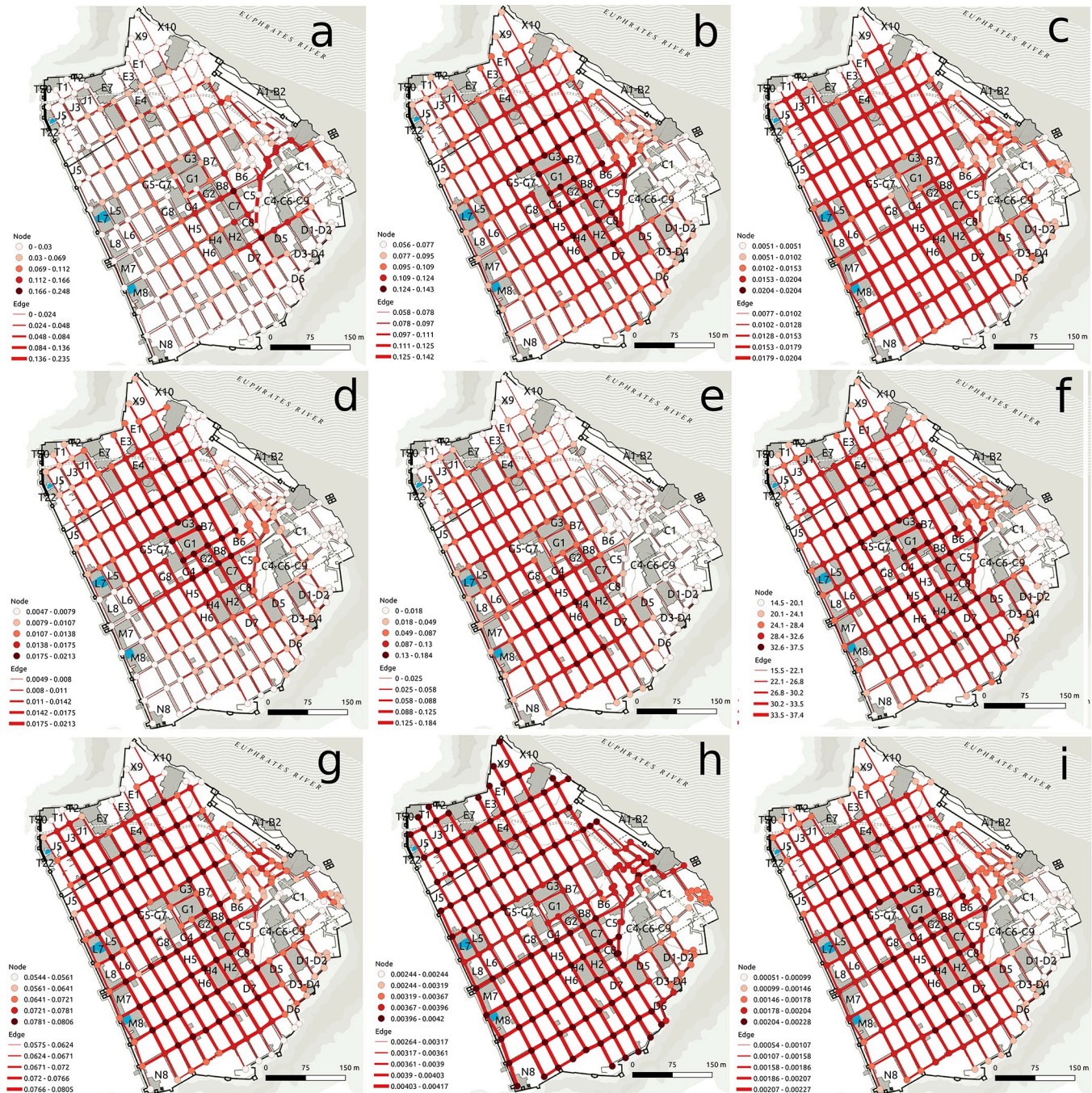

**Fig 2. Geographic representation of centrality measures for streets in the ancient town of Dura Europos in modern Syria.** Centrality measures include betweenness (a), closeness (b), degree (c), efficiency (d), eigenvector (e), harmonic (f), Katz (g), straightness (h), and current flow (i). The background maps used is reprinted under a CC BY 4.0 license with permission from Yale University.

Europos, the measures also show more regions with low centrality values with eigenvector centrality showing highly skewed centrality (Fig 3E) than the other measures. Overall, this demonstrates potential centrality variation in distributions between more orthogonal and less orthogonal urban settings.

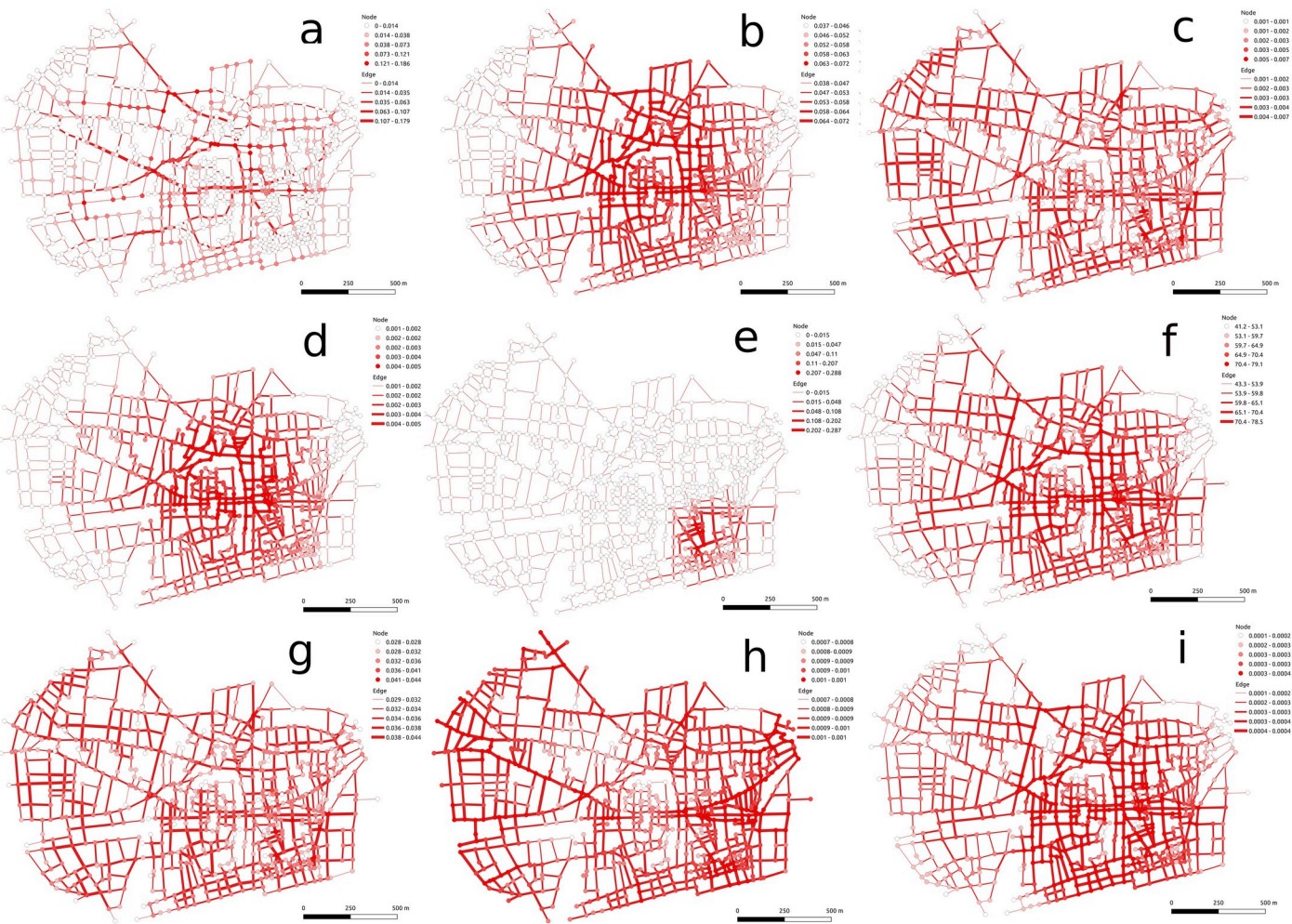

**Fig 3.** Centrality measures include betweenness (a), closeness (b), degree (c), efficiency (d), eigenvector (e), harmonic (f), Katz (g), straightness (h), and current flow (i) shown for streets and nodes for 15th century Barcelona.

The examples above can be extended to look at all urban settings where mostly complete or complete street data are available, which represents 72 samples from the total. Looking at the overall centrality distributions for these sites, and dividing them into orthogonal and hybrid/organic settlement categories, indicates varied patterns demonstrating more even distributions with generally higher overall median centrality for orthogonal urban areas than more hybrid/organic settings (Figs 4 and 5). Additionally, the distributions are compared using a Mann-Whitney-Wilcoxen test with a Holm–Bonferroni method [59], where all distributions, within and between the categories, showed significant differences at p-value<0.01 levels. For Figs 4 and 5, we note that the x- and y-axes are different in the figures due to distribution variations that made the same ranges difficult to apply. In these distributions, urban sites that are more organic generally display more skewness and kurtosis (Table 1), with the exceptions of efficiency and harmonic centrality, as well as mostly lower median centrality values, although harmonic centrality is higher for hybrid/organic settings. For harmonic centrality, values are not normalized, which explains the variation with closeness centrality. Efficiency centrality is generally low for hybrid/organic urban regions, which helps to explain the lower skewness and kurtosis values for this distribution. Overall, the distributions suggest centrality values for orthogonal sites are, based on median values, mostly higher and less skewed based on a variety

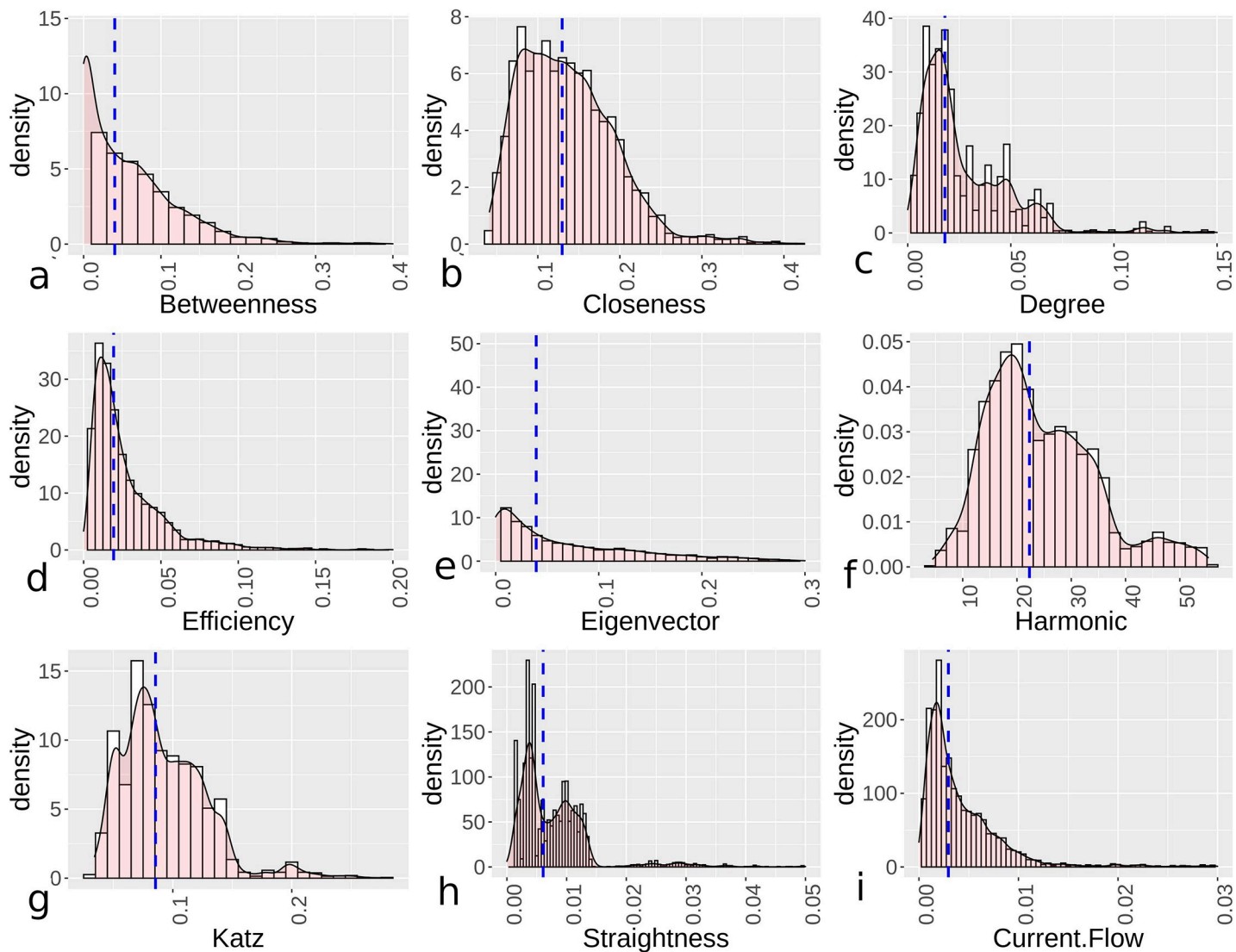

**Fig 4. Kernel density plots showing centrality distributions for orthogonal settlements' centrality values with the dashed lines showing median values.** Graphs depict betweenness (a), closeness (b), degree (c), efficiency (d), eigenvector (e), harmonic (f), Katz (g), straightness (h), and current flow (i).

of distance and degree metrics. From the distributions and using bootstrapping to test distributions [60], hybrid/organic demonstrate more exponential distribution qualities for efficiency, eigenvector, and current flow centrality, while orthogonal towns display more power law relationships for the same centrality measures. Demonstrating variation in eigenvector centrality, Gini coefficient values for nodes in orthogonal sites averaged 0.47, while for hybrid/organic it is 0.71. This demonstrates a wide disparity between lower and upper values for hybrid/organic sites. For other distributions, outside of harmonic, these also display power law qualities if some of the higher values are aggregated.

## Centrality and scaling

Results presented here focus on scaling and centrality measures for datasets that are either complete or likely to be mostly complete urban areas (i.e., 72 urban samples). The number of

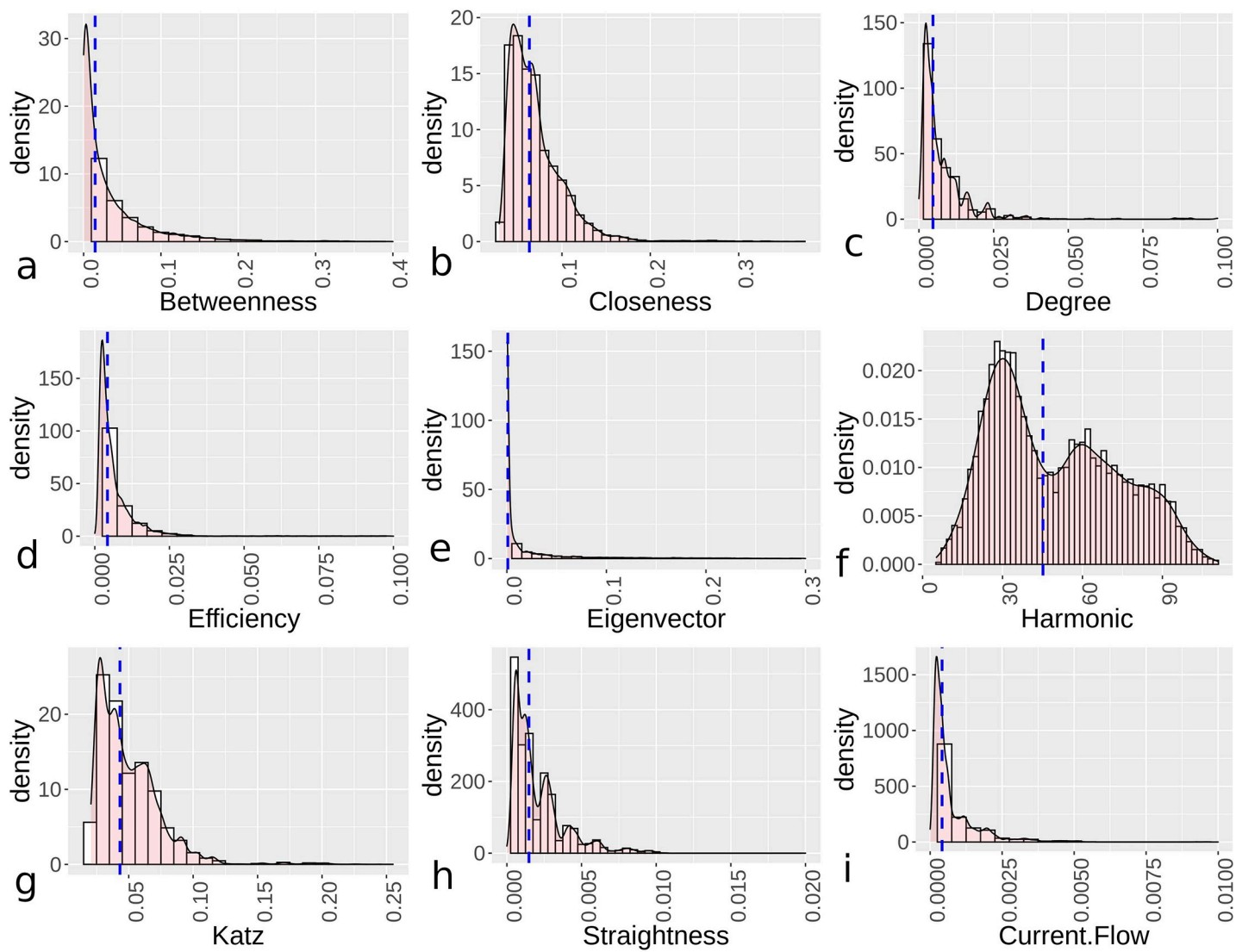

**Fig 5. Kernel density plots showing centrality distributions for hybrid/organic settlements' centrality values with the dashed lines showing median values.** Graphs depict betweenness (a), closeness (b), degree (c), efficiency (d), eigenvector (e), harmonic (f), Katz (g), straightness (h), and current flow (i).

nodes, and node density measured by the number of nodes/area (ha), are scaled to urban area for all sites (Fig 6). Generally, the number and density of nodes over a given area have comparable power law properties, although hybrid/organic urban sites display greater variance. For the most part, hybrid/organic sites have more nodes and greater node density than orthogonal sites. While the overall number of nodes increase in a sub-linear manner as sites are larger, density values have a sub-linear decline for increasing urban areas.

Figs 7 and 8 provide results for orthogonal and hybrid/organic urban sites respectively, applying Eq (2) to site areas and median centrality measures for street networks. The number of sites for periods vary in the two figures due to variation in the types of sites, but outputs show some general comparability. Table 2 provides further summary statistics, with the confidence interval (CI) for $\beta$, mean absolute error (MAE), and root mean squared error (RMSE) for scaling centrality estimates given. Mostly negative area and centrality relationships indicate declining median centrality as sites become larger, comparable to declining node densities.

**Table 1. Results demonstrating skewness and kurtosis for centrality distributions in different settlement types.**

| Measure | Type | Skewness | Kurtosis |
|---|---|---|---|
| Betweenness | Orthogonal | 2.06 | 7.58 |
| Closeness | Orthogonal | 0.94 | 1.4 |
| Degree | Orthogonal | 2.75 | 12 |
| Efficiency | Orthogonal | 21.5 | 487.11 |
| Eigenvector | Orthogonal | 1.47 | 2.21 |
| Harmonic | Orthogonal | 0.78 | 0.3 |
| Katz | Orthogonal | 1.18 | 2.28 |
| Straightness | Orthogonal | 2.81 | 11.45 |
| Current Flow | Orthogonal | 2.94 | 12.43 |
| Betweenness | Hybrid/Organic | 3.28 | 15.76 |
| Closeness | Hybrid/Organic | 2.56 | 12.19 |
| Degree | Hybrid/Organic | 6.15 | 59.42 |
| Efficiency | Hybrid/Organic | 9.14 | 136.15 |
| Eigenvector | Hybrid/Organic | 3.7 | 16.81 |
| Harmonic | Hybrid/Organic | 0.4 | -0.93 |
| Katz | Hybrid/Organic | 2.08 | 8.3 |
| Straightness | Hybrid/Organic | 6.2 | 54.07 |
| Current Flow | Hybrid/Organic | 8.13 | 82.85 |

Orthogonal urban locations generally show higher sub-linear $\beta$ results than hybrid/organic sites, with closeness centrality showing more comparable results in the two categories. Eigenvector centrality, particularly for hybrid/organic sites, showed wide disparity overall. In fact, hybrid/organic sites generally had greater variance in the relationship between median centrality values and area measures.

## Incomplete street data

In many cases, archaeological sites are often only partially explored. Here, we attempt to understand if partially excavated sites, which included 17 of the total 89 sites studied (4 hybrid/organic and 13 orthogonal), could be used and yield centrality results comparable to orthogonal and hybrid/organic urban sites. In this case, since samples are limited, orthogonal and hybrid/organic are combined as one dataset. The number of nodes and node density show somewhat comparable β values to that of more complete sites, although the number of nodes scaled at a lower level and density had a more negative relationship to area than more complete sites (Fig 9). We also applied centrality values for sites, using only the areas and streets uncovered and incorporating this partial data into the scaling method represented by Eq (2). Fig 10 provides centrality results comparable to the orthogonal and hybrid/organic results presented earlier. Similar to these sites, it is evident that centrality scaling is somewhat similar, even with a relatively limited sample set or incomplete road networks. Overall, the centrality values all fall within CI ranges provided in Table 2 for β values. Similar to the previous centrality results, eigenvector and current flow centrality demonstrate relatively greater variability.

## Discussion

### Benefits and key results

This work has presented an approach that investigates street network centrality and population scaling. Data from ancient and early Modern urban sites in Europe, the Middle East, and

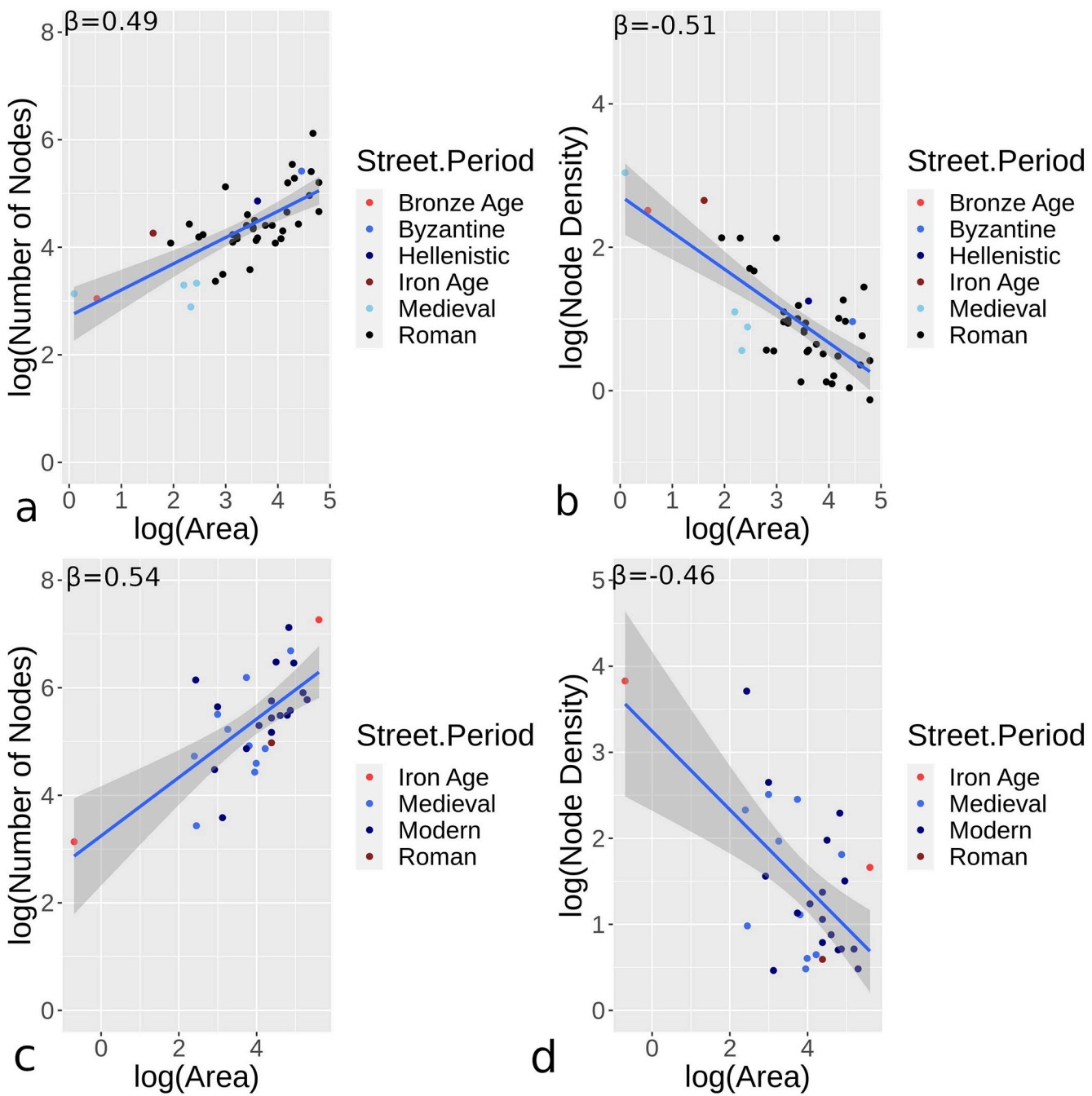

**Fig 6.** Orthogonal (a-b) and hybrid/organic (c-d) urban sites studied with the number of nodes (a,c) and density (number of nodes/ha; b,d) compared to area.

North Africa are collected, where results demonstrate comparable scaling values for the varied periods and regions. As a wider contribution, this work presents the street centrality tool as an open contribution along with street data and centrality outputs presented. Structurally, some centrality distribution measures show some similarity between orthogonal and hybrid/organic urban sites, but there are evident differences between orthogonal and hybrid/organic sites. All distributions showed significant differences when statistically compared. OOrthogonal sites

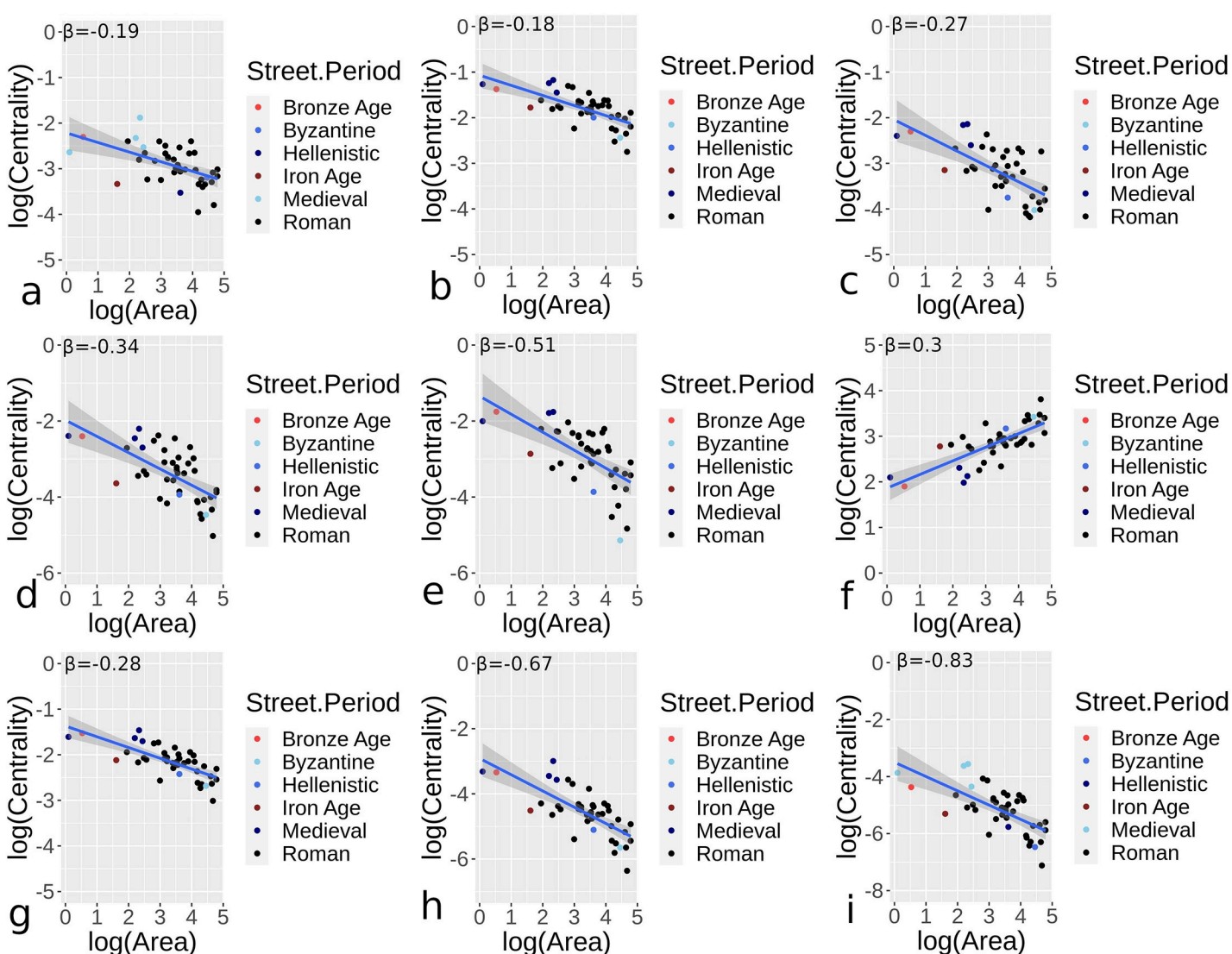

**Fig 7. Centrality and area scaling values for urban settings in different periods and for all nearly complete or complete street networks for orthogonal street networks.** Results demonstrate betweenness (a), closeness (b), degree (c), efficiency (d), eigenvector (e), harmonic (f), Katz (g), straightness (h), and current flow (i) centrality values.

generally had higher overall median centrality values, both for degree-based and distance-based measures. There is a greater tendency for hybrid/organic sites to skew towards lower centrality values. Among measures, this is most evident in eigenvector and current flow centrality. In measures that normalize for the number of nodes, we see such measures as betweenness and straightness centrality generally lower for hybrid/organic urban sites. A limited number of measures, including efficiency, eigenvector, and current flow centrality, demonstrate exponential distributions for hybrid/organic urban locations, whereas in modern systems exponential distributions were found for planned urban sites (i.e., orthogonal cities) [3]. Harmonic centrality displayed distributions that are the most similar to normal distributions, where the measure looked at centrality based on how close other nodes were, but for this normalization is not applied. In fact, harmonic, degree, straightness, and Katz centrality distributions for orthogonal cities also appear somewhat similar to bi-modal. This all indicates some

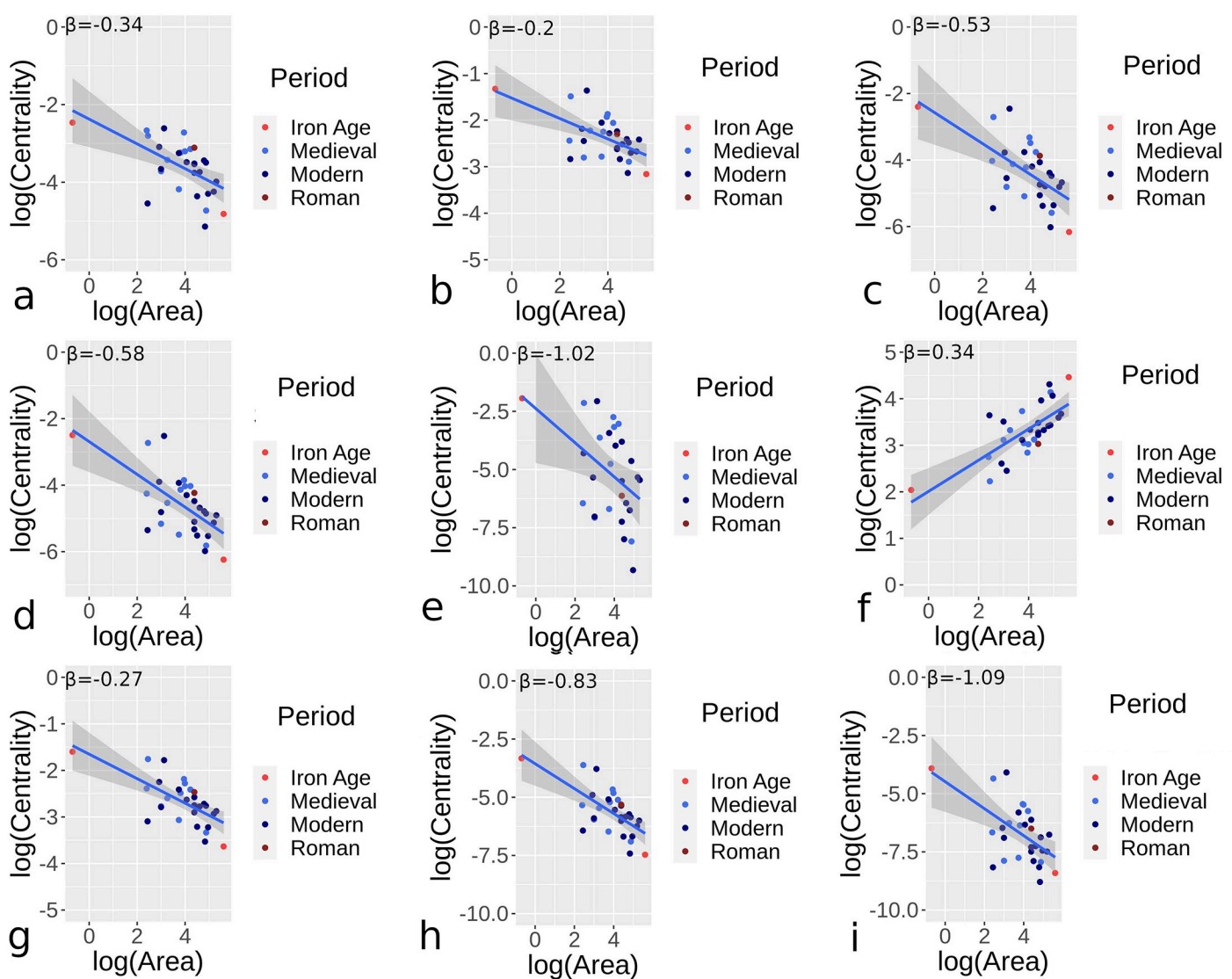

**Fig 8. Centrality and area scaling values for urban settings in different periods and for all nearly complete or complete street networks for hybrid/organic street networks.** Results demonstrate betweenness (a), closeness (b), degree (c), efficiency (d), eigenvector (e), harmonic (f), Katz (g), straightness (h), and current flow (i) centrality values.

evenness in the results for orthogonal streets, where there was more evenness in connectivity and distances. Overall, results suggest that shape and configuration of street networks have noticeable effects on how well urban locations connect to different locations and neighborhoods. Orthogonal sites are suggested to be better at connecting a wider area of urban sites with higher overall median node centrality for measures. Such results are comparable to modern cities, which have shown similar centrality differences in comparing orthogonal and hybrid/organic settings, particularly when comparable modes of transport are accounted for [61, 62].

The number of nodes and node density have positive and negative sub-linear relationships respectively for both orthogonal and hybrid/organic sites. Most of the results for median centrality measures and urban areas demonstrate power-law relationships. Centrality values display mostly sub-linear growth, with eigenvector and current flow centrality displaying β<-1.0

**Table 2. Summary values showing *β*, CI for *β*, and MAE and RMSE for centrality/reciprocal centrality for orthogonal and hybrid/organic urban sites.**

| Centrality | Type | Exponent (β) | Exponent (β) (95% CI) | prefactor (Y0) | MAE | RMSE |
|---|---|---|---|---|---|---|
| Betweenness | Orthogonal | -0.19 | -0.28 to—0.06 | 0.1 | 0.02 | 0.02 |
| Closeness | Orthogonal | -0.18 | -0.27 to -0.10 | 0.3 | 0.03 | 0.04 |
| Degree | Orthogonal | -0.27 | -0.40 to -0.11 | 0.1 | 0.02 | 0.02 |
| Efficiency | Orthogonal | -0.34 | -0.48 to -0.13 | 0.1 | 0.02 | 0.02 |
| Eigenvector | Orthogonal | -0.51 | -0.73 to -0.33 | 0.3 | 0.03 | 0.04 |
| Harmonic | Orthogonal | 0.3 | 0.21 to 0.39 | 6.4 | 4.08 | 5.37 |
| Katz | Orthogonal | -0.28 | -0.37 to -0.22 | 0.3 | 0.03 | 0.03 |
| Straightness | Orthogonal | -0.67 | -0.87 to -0.54 | 0.1 | 0.01 | 0.01 |
| Current Flow | Orthogonal | -0.83 | -1.14 to -0.7 | 0.1 | 0.01 | 0.01 |
| Betweenness | Hybrid/Organic | -0.34 | -0.54 to -0.12 | 0.1 | 0.01 | 0.02 |
| Closeness | Hybrid/Organic | -0.2 | -0.33 to -0.05 | 0.2 | 0.03 | 0.04 |
| Degree | Hybrid/Organic | -0.53 | -0.84 to -0.27 | 0.1 | 0.01 | 0.02 |
| Efficiency | Hybrid/Organic | -0.58 | -0.90 to -0.36 | 0.1 | 0.01 | 0.02 |
| Eigenvector | Hybrid/Organic | -1.02 | -1.80 to -0.22 | 0.2 | 0.03 | 0.06 |
| Harmonic | Hybrid/Organic | 0.34 | 0.19–0.49 | 7.3 | 10.5 | 14.04 |
| Katz | Hybrid/Organic | -0.27 | -0.42 to -0.15 | 0.2 | 0.02 | 0.03 |
| Straightness | Hybrid/Organic | -0.83 | -1.25 to -0.63 | 0.1 | 0.01 | 0.03 |
| Current Flow | Hybrid/Organic | -1.09 | -1.74 to -0.84 | 0.1 | 0.01 | 0.04 |

for hybrid/organic urban sites. In fact, there is wide disparity for eigenvector and current flow for these types of sites, suggesting these measures are less effective in demonstrating clear centrality and scaling relationships compared to other methods. For these measures, the drop off

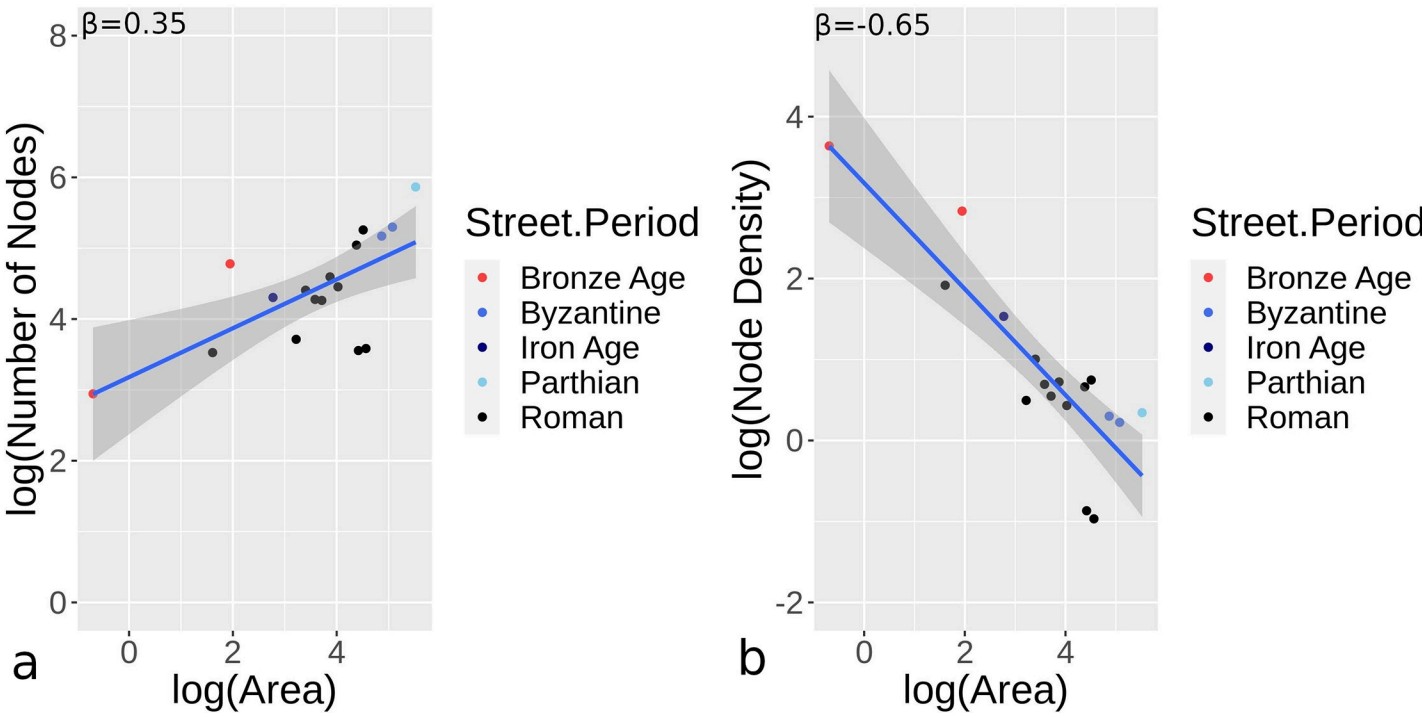

**Fig 9.** The scaling relationship for the number of nodes (a) and density (number of nodes/ha; b) for incomplete street network sites.

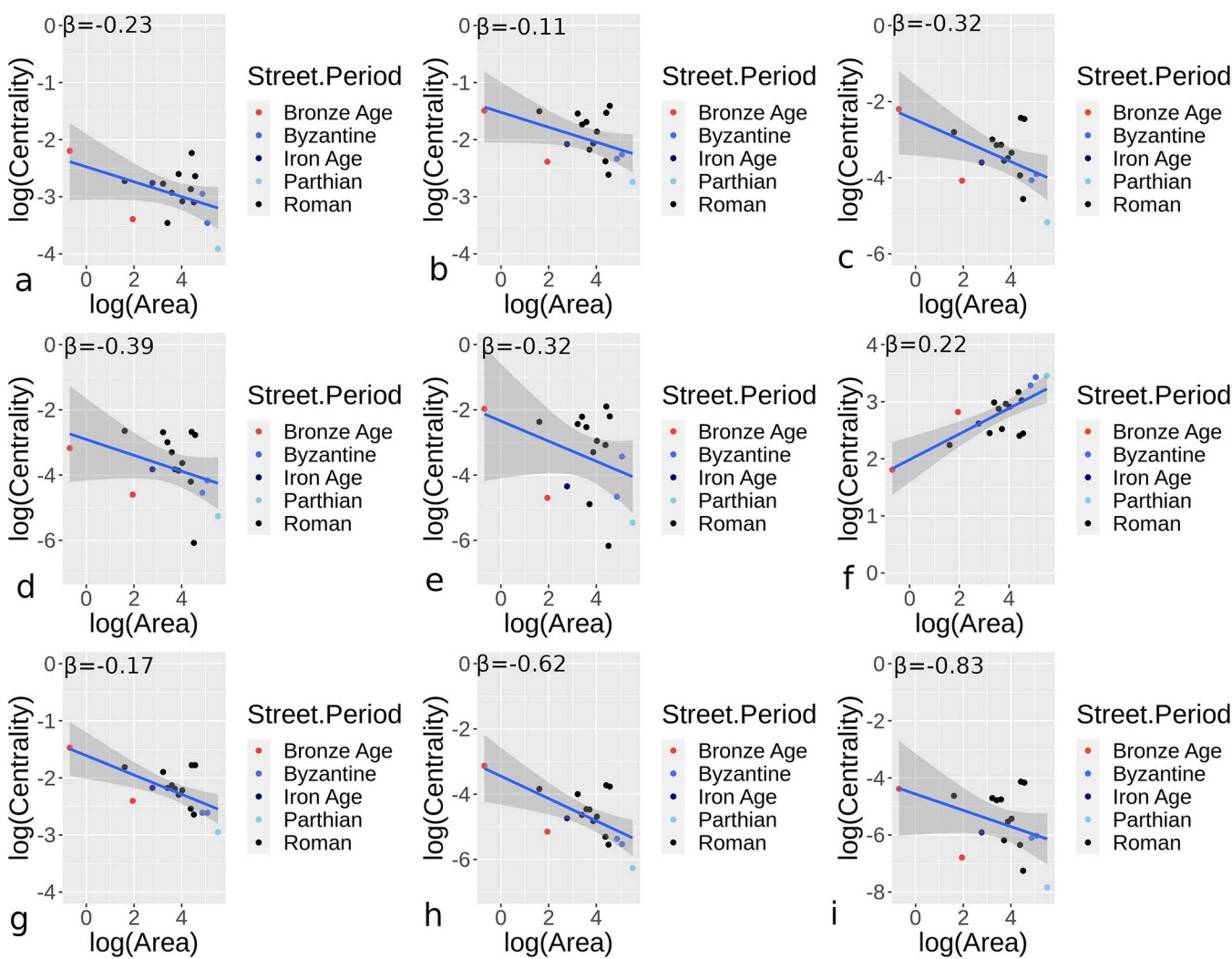

**Fig 10. Centrality and scaling values for incomplete street networks for orthogonal and hybrid/organic sites.** Results demonstrate betweenness (a), closeness (b), degree (c), efficiency (d), eigenvector (e), harmonic (f), Katz (g), straightness (h), and current flow (i) centrality values.

in median centrality values is greater than the rate of urban growth. Overall for hybrid/organic sites, at least when evaluating median centrality values, these values more rapidly diminish as urban centers become larger. This suggests that hybrid/organic sites not only show lower centrality values but that centrality declines more rapidly than in orthogonal urban settings as urban centers become larger. We also see that for incomplete urban locations, node numbers, node density, and centrality and area scaling are also comparable to complete sites. This suggests that there is a potential to estimate different median centrality values for sites, using the prefactors and β range estimates determined (Table 2), where street data are more difficult to obtain, particularly in archaeological cases or sites with incomplete exposure.

Although each measure looks at centrality differently, the results demonstrate dissipation of interactions that occur, something observed for modern urban infrastructure and districts in measuring power law relationships and population [43]. This has implications that larger cities diminish abilities to interact across a wider urban environment and could limit overall social interaction and urban growth. Such results are not surprising as a greater population creates

not only more traffic but also distances across urban areas increase, making it more difficult to reach desired areas. In effect, diminished centrality reflects a cost to urban growth, where diminished social benefit possibilities can serve as limitations to urban area growth [63]. Nevertheless, as orthogonal urban sites appear to create greater median centrality scores as sites become larger, this could have implications for social interactions and activities. Specifically, greater centrality values have been associated with areas demonstrating greater opportunities for social interaction across wider urban areas in the past and present [22, 23, 26]. One possibility is that orthogonal sites may better encourage social interaction or at least diminish some of the costs of larger urban growth as suggested by higher median centrality values and scaling results. However, we note that the analysis carried out neither accounted for modes of transport nor assessment of how traffic would have been regulated in any one urban site. For instance, it is possible pedestrian and animal-based traffic could have yielded different centrality results if factors such as traffic and width of streets are accounted for. Other research has indicated that when accounting for varied modes of transport and factors affecting them, then street centrality is not only different for the same locations but different transport choices could be made based on how streets are spatially organized [43, 64]. From our results, we suggest that there is potential for orthogonal sites to better facilitate social interaction over wider areas than hybrid/organic sites.

One result that may have wider theoretical implications is that the centrality scaling relationships demonstrate $\beta$ ratios that are comparable to other forms of urban infrastructure, such as the widths of city gates and incoming inter-urban road networks [9, 31, 33]. In those cases, β is positive, rather than negative, but the ratio of growth is comparable, with exponents being at around ⅙, ⅓ (or multiples of these values). This suggests that the internal and external properties of intra- and inter-urban transportation networks grow in a similar manner, relative to size and population, as urban regions extend. Similarity in power-law relationships and centrality distributions for street networks are evident in modern and pre-industrial cities, although shapes of cities appear to have different results between more recent and pre-industrial urban sites [3]. This could be explained perhaps by the effects of more modern transport on cities relative to pre-industrial sites, although this cannot be stated for certain without further investigation.

## Conclusion

### Limitations and future research

Results are limited by the fact that only 89 sites are used. Centrality distributions have some comparability with modern urban contexts [3], suggesting that increasing the dataset may not substantially change some of the general trends indicated here. We recognize, as we have combined street data, this has meant we have ignored temporal variation within given periods, as potentially not all streets within an urban context were actively used contemporaneously as other streets. Furthermore, traffic flows and transport modes are not analyzed, where our research focused on centrality potential and scaling.

For future work, we see that results could be broken down into regions, identifying where patterns are comparable or similarities are evident. In particular, expanding data across wider regions outside of mainly Europe and the Middle East would potentially allow us to demonstrate if properties observed here are likely to be common across other regions. Additionally, connecting scaling properties between street networks and other urban infrastructure is possible, given that the scaling exponent demonstrates properties similar to other infrastructure studied [31]. In such cases, there could be broad similarity between urban infrastructure and population that indicate such features demonstrate comparable power law relationships. This

result would be worth following up, particularly for urban sites where historical population data could be compared with given street networks. Furthermore, studies have also demonstrated that varied centrality scores correlate with specialized and different social activities [65, 66]. It may be possible to expand this work for sites to see how specific activities or sites/buildings, such as commercial, leisure, religious, or other activities, demonstrate varied relationships to centrality scores, particularly when varied modes of transport and traffic are accounted for. How ancient traffic and mobility within urban environments interact can be studied using network methods such as that proposed here, while then expanding to how population scaling affects this. However, results should be combined with fieldwork methods that can assist in validating relative traffic patterns assessment. Robustness and resilience of street networks, through modeled removal of nodes in network analysis, could also be investigated using comparable methods presented. We see that there are many potential avenues of future research given results observed; our endeavor has been to enable more complete urban street analysis to be possible by aggregating datasets, making them available, and providing methods that enable continued work in this area.

## Supporting information

**S1 File.**
(DOCX)

## Acknowledgments

We would like to thank the reviewers for their useful comments and suggestions.

## Author Contributions

**Conceptualization:** Mark Altaweel, Jack Hanson, Andrea Squitieri.

**Data curation:** Mark Altaweel, Jack Hanson, Andrea Squitieri.

**Formal analysis:** Mark Altaweel, Andrea Squitieri.

**Funding acquisition:** Mark Altaweel.

**Investigation:** Mark Altaweel, Jack Hanson.

**Methodology:** Mark Altaweel, Jack Hanson, Andrea Squitieri.

**Project administration:** Mark Altaweel.

**Resources:** Mark Altaweel.

**Software:** Mark Altaweel.

**Validation:** Mark Altaweel.

**Visualization:** Mark Altaweel.

**Writing – original draft:** Mark Altaweel, Jack Hanson, Andrea Squitieri.

**Writing – review & editing:** Mark Altaweel, Jack Hanson, Andrea Squitieri.

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
