## [Decision Letter · Decision Letter 0]

16 Aug 2021

PONE-D-21-15963

The structure, centrality, and scale of urban street networks: Cases from Pre-Industrial Afro-Eurasia

PLOS ONE

Dear Dr. Altaweel,

Thank you for submitting your manuscript to PLOS ONE. After careful consideration, we feel that it has merit but does not fully meet PLOS ONE’s publication criteria as it currently stands. Therefore, we invite you to submit a revised version of the manuscript that addresses the points raised during the review process.

In particular, please address with attention and details the comments requiring further comparisons on the different indicators presented, that might be presenting similar, correlated information, and the clarification of the manuscript's text in some of its passages.

We look forward to receiving your revised manuscript.

Kind regards,

Riccardo Gallotti

Academic Editor

PLOS ONE

1. Please ensure that your manuscript meets PLOS ONE's style requirements, including those for file naming. The PLOS ONE style templates can be found at https://journals.plos.org/plosone/s/file?id=wjVg/PLOSOne_formatting_sample_main_body.pdf and https://journals.plos.org/plosone/s/file?id=ba62/PLOSOne_formatting_sample_title_authors_affiliations.pdf.

2. In your manuscript, please provide additional information regarding the specimens used in your study. Ensure that you have reported specimen numbers and complete repository information, including museum name and geographic location.

For more information on PLOS ONE's requirements for paleontology and archaeology research, see https://journals.plos.org/plosone/s/submission-guidelines#loc-paleontology-and-archaeology-research.

5. We note that Figures 1-3 in your submission contain [map/satellite] images which may be copyrighted. All PLOS content is published under the Creative Commons Attribution License (CC BY 4.0), which means that the manuscript, images, and Supporting Information files will be freely available online, and any third party is permitted to access, download, copy, distribute, and use these materials in any way, even commercially, with proper attribution. For these reasons, we cannot publish previously copyrighted maps or satellite images created using proprietary data, such as Google software (Google Maps, Street View, and Earth). For more information, see our copyright guidelines: http://journals.plos.org/plosone/s/licenses-and-copyright.

a. You may seek permission from the original copyright holder of Figures 1-3 to publish the content specifically under the CC BY 4.0 license. 

Additional Editor Comments (if provided):

Reviewers' comments:

Reviewer's Responses to Questions

**Comments to the Author**

1. Is the manuscript technically sound, and do the data support the conclusions?

Reviewer #1: Yes

Reviewer #2: Yes

2. Has the statistical analysis been performed appropriately and rigorously? 

Reviewer #1: Yes

Reviewer #2: Yes

3. Have the authors made all data underlying the findings in their manuscript fully available?

Reviewer #1: Yes

Reviewer #2: Yes

4. Is the manuscript presented in an intelligible fashion and written in standard English?

Reviewer #1: Yes

Reviewer #2: Yes

5. Review Comments to the Author

Reviewer #1: Title

The structure, centrality, and scale of urban street networks: Cases from Pre-Industrial Afro-Eurasia

This ms seeks to describe scaling relationships and Centrality of urban street networks across a large set of pre-intdustrial and ancient cities in Eurasia.

I enjoyed reading this ms. It is well written and scientifically sound. Methods are wel described.

This is a great contribution to the literature building on specific findings of urban scaling.

I have some concerns regarding the definition of hybrid/organic cities, while it seem intuitive, it would be helpfull to see which cities have been classified as such. A suggestion for this would be to include such information in Figure 1 perhaps.

My next concern regards the analyses of density plots. Some Centrality metrics show bimodal distribution. Why is that? Could this be related to my previous point? Particularly, Harmonic, Degree, Straightness, Katz and Betweenness to some degree. This is worth mentioning in the discussion.

Reviewer #2: General comment

The paper shows results of common centrality properties for different types of cities found in different regions as well as different periods.

It's a very interesting idea to investigate scaling relationships for centrality measures in different types of cities. The methodology and the outputs are also very interesting with a significant added value within the geo-archeological fieldwork.

The results demonstrated different centrality distributions and different scaling relationships between planned and organic cities.

Minor global comments:

- There are many indicators computed and presented. One could argue that some of them may bring the same kind of information. It could be interesting to show that they are not giving correlated results.

- The text is overall very well structured and clear, but few times sentences are quite long, with a complex structure, and we miss the point. Those specific sentences are detailed in the next paragraph “detailed comments.”

Detailed comments:

Abstract

l.20 – l.22: The sentence is hard to read. It would be better to reformulate.

Introduction

l. 59: The term of “urban area” could be better defined.

l. 69: repetition of “assess”

l.79: “…and centrality and urban area.” You can use a comma to avoid repeating “and” and make the sentence clearer.

l.80: We have the information about the layouts after but few words more about it in this part could be relevant.

l. 97: repetition of “how”

l.112 – l.116: The sentence is too complex. It would be better to simplify by writing one sentence for one idea at a time.

Background

The background section presents the scientific context of the paper that was already addressed in the introduction section without being detailed. It seems possible to avoid some redundancies between the two sections.

Urban Scaling

In this paragraph the word “demonstrate” is used 12 times! It would be better to reformulate. (Especially the sentence l. 131 to 134).

l. 134 : typo : [7,30)

Materials and Methods

l. 173-174: The classification in “orthogonal” or “organic” would gain to be developed

l.229 – l.233: The sentence is unclear (too long).

l.247: I would suggest adding one sentence to explain the 30 degree threshold.

Results

- In figures 4 and 5, x and y axes are not with the same scale between plots of the same indicator for each types, which make their comparison harder.

- In figure 5 the legend mention a “dashed lines showing median values” which is not present on the plots.

- In figure 6, same problem of scale between 6a and 6c on the one hand, and 6b and 6d on the other.

- Figures 7 and 8: Why it is not the same number of periods? Is it related to data availability? It would be better to explain why the results are comparable while the inputs considered are not the same for the two different street layout type.

6. PLOS authors have the option to publish the peer review history of their article (what does this mean?). If published, this will include your full peer review and any attached files.

Reviewer #1: **Yes: **Horacio Samaniego

Reviewer #2: **Yes: **Claire Lagesse

---

## [Author Response · Author response to Decision Letter 0]

24 Aug 2021

Dear Riccardo,

We very much appreciate this review and have accepted nearly all the comments indicated to us. We are happy to revise and provide the following list of comments, indicated in red for our response to specific editorial or review comments, in this letter. We also indicate the letter provided to us as a reference and our comments are embedded. Where we did not address a given concern, we indicate why and provide justification. However, in general, we have accepted comments given to us and address them. Thank you again and we appreciate this review.

Comments:

PONE-D-21-15963

The structure, centrality, and scale of urban street networks: Cases from Pre-Industrial Afro-Eurasia

PLOS ONE

Dear Dr. Altaweel,

Thank you for submitting your manuscript to PLOS ONE. After careful consideration, we feel that it has merit but does not fully meet PLOS ONE’s publication criteria as it currently stands. Therefore, we invite you to submit a revised version of the manuscript that addresses the points raised during the review process.

We have followed the editorial comments and reviewer comments. We hope these are adequately addressed. We indicate how we addressed points below under each point indicated.

In particular, please address with attention and details the comments requiring further comparisons on the different indicators presented, that might be presenting similar, correlated information, and the clarification of the manuscript's text in some of its passages.

We changed the financial disclosure in the Cover Letter, indicating the funding source of this work.

We look forward to receiving your revised manuscript.

Kind regards,

Riccardo Gallotti

Academic Editor

PLOS ONE

1. Please ensure that your manuscript meets PLOS ONE's style requirements, including those for file naming. The PLOS ONE style templates can be found at https://journals.plos.org/plosone/s/file?id=wjVg/PLOSOne_formatting_sample_main_body.pdf and https://journals.plos.org/plosone/s/file?id=ba62/PLOSOne_formatting_sample_title_authors_affiliations.pdf.

We checked and followed the guidelines provided.

2. In your manuscript, please provide additional information regarding the specimens used in your study. Ensure that you have reported specimen numbers and complete repository information, including museum name and geographic location.

We provided a .shp file for the location information. The data are all referenced, with the relevant shapefiles provided which derived from the maps referenced in the Supplementary Material repository provided. All data are in the Supplementary Materail link provided.

For more information on PLOS ONE's requirements for paleontology and archaeology research, see https://journals.plos.org/plosone/s/submission-guidelines#loc-paleontology-and-archaeology-research.

We have now provided an indication in the article in the Case Studies section discussing that the information used required no permissions as they were all public and/or published data.

The Center of Advanced Studies-Schwerpunkt (Siedlungen zwischen Diversität und Homogenität) from the University of Munich (LMU) funded this work through a fellowship. There is no number of the award as it funded a sabbatical leave when the work was developed. We indicated this in the Acknowledgments and Funding Information now. We hope it is clear.

The billing should go to UCL, which is the first author’s affiliation. They have an agreement/payment setup with PlosOne for articles to be published. In the revision, we do not see where to amend this or indicate?

5. We note that Figures 1-3 in your submission contain [map/satellite] images which may be copyrighted. All PLOS content is published under the Creative Commons Attribution License (CC BY 4.0), which means that the manuscript, images, and Supporting Information files will be freely available online, and any third party is permitted to access, download, copy, distribute, and use these materials in any way, even commercially, with proper attribution. For these reasons, we cannot publish previously copyrighted maps or satellite images created using proprietary data, such as Google software (Google Maps, Street View, and Earth). For more information, see our copyright guidelines: http://journals.plos.org/plosone/s/licenses-and-copyright.

We have removed the underlying maps for Figures 1 and 3. We have now added Natural Earth data for Figure 1’s underlying data and the credit is given in the caption. For Figure 2, we now have permission to use the image and this is indicated in the caption and uploaded permission document.

a. You may seek permission from the original copyright holder of Figures 1-3 to publish the content specifically under the CC BY 4.0 license. 

We have added four new references. These references are numbed 37-39 and 59 in the references given.

Additional Editor Comments (if provided):

Reviewers' comments:

Reviewer's Responses to Questions

Comments to the Author

1. Is the manuscript technically sound, and do the data support the conclusions?

Reviewer #1: Yes

Reviewer #2: Yes

2. Has the statistical analysis been performed appropriately and rigorously? 

Reviewer #1: Yes

Reviewer #2: Yes

3. Have the authors made all data underlying the findings in their manuscript fully available?

Reviewer #1: Yes

Reviewer #2: Yes

4. Is the manuscript presented in an intelligible fashion and written in standard English?

Reviewer #1: Yes

Reviewer #2: Yes

5. Review Comments to the Author

Reviewer #1: Title

The structure, centrality, and scale of urban street networks: Cases from Pre-Industrial Afro-Eurasia

This ms seeks to describe scaling relationships and Centrality of urban street networks across a large set of pre-intdustrial and ancient cities in Eurasia.

I enjoyed reading this ms. It is well written and scientifically sound. Methods are wel described.

This is a great contribution to the literature building on specific findings of urban scaling.

I have some concerns regarding the definition of hybrid/organic cities, while it seem intuitive, it would be helpfull to see which cities have been classified as such. A suggestion for this would be to include such information in Figure 1 perhaps.

We have now put which cities were classified as ‘orthogonal’, ‘organic’ and ‘hybrid’ in Figure 1. We also provided definitions of the categories used in the beginning of the material and methods section. The data in the Supplementary Material also indicates what urban locations are what type.

My next concern regards the analyses of density plots. Some Centrality metrics show bimodal distribution. Why is that? Could this be related to my previous point? Particularly, Harmonic, Degree, Straightness, Katz and Betweenness to some degree. This is worth mentioning in the discussion.

This is now discussed in the discussion section. It appears that this is the case mostly for orthogonal, which we believe relates to the fact that such cities promote somewhat higher connectivity across wider areas of urban layouts than organic or even hybrid sites. So what is going on is that the distributions are more spread, and somewhat bimodal in places, relative to organic/hybrid urban locations. We explain this in the discussion section.

Reviewer #2: General comment

The paper shows results of common centrality properties for different types of cities found in different regions as well as different periods.

It's a very interesting idea to investigate scaling relationships for centrality measures in different types of cities. The methodology and the outputs are also very interesting with a significant added value within the geo-archeological fieldwork.

The results demonstrated different centrality distributions and different scaling relationships between planned and organic cities.

Minor global comments:

- There are many indicators computed and presented. One could argue that some of them may bring the same kind of information. It could be interesting to show that they are not giving correlated results.

We provide a statistical comparison in the results. We indicate that at a p-value<0.01 significance level, all the distributions (between and within each category) are significantly different. We were also a little unclear about this comment but we hope we understood it and addressed it appropriately. We also qualitatively describe the distributions in the results and discussion.

- The text is overall very well structured and clear, but few times sentences are quite long, with a complex structure, and we miss the point. Those specific sentences are detailed in the next paragraph “detailed comments.”

We appreciate these suggestions and we have mostly followed them.

Detailed comments:

Abstract

l.20 – l.22: The sentence is hard to read. It would be better to reformulate.

This is now changed and the sentence is edited to be more clear.

Introduction

l. 59: The term of “urban area” could be better defined.

This is now defined (briefly) in the introduction.

l. 69: repetition of “assess”

Many of these are removed and other words are now used.

l.79: “…and centrality and urban area.” You can use a comma to avoid repeating “and” and make the sentence clearer.

We revised to make it clearer.

l.80: We have the information about the layouts after but few words more about it in this part could be relevant.

Ok a fuller definition is now provided in the material and methods section, but an indication of the types of settlements evaluated is now provided in the introduction as well.

l. 97: repetition of “how”

This is now edited.

l.112 – l.116: The sentence is too complex. It would be better to simplify by writing one sentence for one idea at a time.

This is now edited.

Background

The background section presents the scientific context of the paper that was already addressed in the introduction section without being detailed. It seems possible to avoid some redundancies between the two sections.

We removed material that was repetitive or elaborated on points in the background that were indicated in the introduction.

Urban Scaling

In this paragraph the word “demonstrate” is used 12 times! It would be better to reformulate. (Especially the sentence l. 131 to 134).

l. 134 : typo : [7,30)

Good suggestions! We have reformed this section and followed the suggestions.

Materials and Methods

l. 173-174: The classification in “orthogonal” or “organic” would gain to be developed

We have now added a paragraph at the beginning of the section to clarify this.

l.229 – l.233: The sentence is unclear (too long).

This is now edited.

l.247: I would suggest adding one sentence to explain the 30 degree threshold.

We have now added an explanation. We indicate that the degree is sharp enough where it represents an intersection.

Results

- In figures 4 and 5, x and y axes are not with the same scale between plots of the same indicator for each types, which make their comparison harder.

Yes, we did try to make the scales the same but due to distribution differences it was not feasible since some of the information, particularly in figure 5, would be hard to notice or see. We have now stated this in the results to make this clear.

- In figure 5 the legend mention a “dashed lines showing median values” which is not present on the plots.

This is now changed so that there is a dashed line for Figure 5.

- In figure 6, same problem of scale between 6a and 6c on the one hand, and 6b and 6d on the other.

Yes this is also due to variations in where values concentrated, which was different between 6a-6d. We tried to standardize but it made some of the values hard to see, which is why we made the outputs then with somewhat different scales. The trends are hopefully clear.

- Figures 7 and 8: Why it is not the same number of periods? Is it related to data availability? It would be better to explain why the results are comparable while the inputs considered are not the same for the two different street layout type.

Yes, there are variations in how much data for each period and also the types of cities (orthogonal vs. organic/hybrid). However, some comparability is evident since we get some similar results. We explain this now in the results section for the figures.

6. PLOS authors have the option to publish the peer review history of their article (what does this mean?). If published, this will include your full peer review and any attached files.

Yes this is fine to publish the history of the peer review.

Do you want your identity to be public for this peer review? For information about this choice, including consent withdrawal, please see our Privacy Policy.

Reviewer #1: Yes: Horacio Samaniego

Reviewer #2: Yes: Claire Lagesse

We have checked our figures using this site and all of them appear to be of good quality and sufficient. Also, we noticed in the notes in the pdf of the manuscript it stated that the colors were hard to read for some figures. However, when looking at the high quality tiff files, the colors look very clearly different. We think that the pdf diminished the quality of the figures, which may explain why the colors are hard to differentiate.

---

## [Decision Letter · Decision Letter 1]

25 Oct 2021

The structure, centrality, and scale of urban street networks: Cases from Pre-Industrial Afro-Eurasia

PONE-D-21-15963R1

Dear Dr. Altaweel,

We’re pleased to inform you that your manuscript has been judged scientifically suitable for publication and will be formally accepted for publication once it meets all outstanding technical requirements. In this process, I recommend to also apply the corrections to the text suggested by reviewer 2.

Kind regards,

Riccardo Gallotti

Academic Editor

PLOS ONE

Additional Editor Comments (optional):

Reviewers' comments:

Reviewer's Responses to Questions

**Comments to the Author**

1. If the authors have adequately addressed your comments raised in a previous round of review and you feel that this manuscript is now acceptable for publication, you may indicate that here to bypass the “Comments to the Author” section, enter your conflict of interest statement in the “Confidential to Editor” section, and submit your "Accept" recommendation.

Reviewer #1: All comments have been addressed

Reviewer #2: All comments have been addressed

2. Is the manuscript technically sound, and do the data support the conclusions?

Reviewer #1: Yes

Reviewer #2: Yes

3. Has the statistical analysis been performed appropriately and rigorously? 

Reviewer #1: Yes

Reviewer #2: Yes

4. Have the authors made all data underlying the findings in their manuscript fully available?

Reviewer #1: Yes

Reviewer #2: Yes

5. Is the manuscript presented in an intelligible fashion and written in standard English?

Reviewer #1: Yes

Reviewer #2: Yes

6. Review Comments to the Author

Reviewer #1: I am very happy with the way that authors have addressed my concerns and now see a clear improvement in the article quality. I congratulate the authors and recommend the publication of this manuscript.

Reviewer #2: Here is a list of minor comments about the text. Please note that the absence of line numbering gives the references to the text not very handy.

Abstract :

“This article presents an approach that investigates past street network centrality measures urban and its relationship to population scaling in urban context”.

: typo (the word "urban" should be deleted).

Introduction :

“Settlement scaling approaches have been used in archaeology as a means to better understand how a variety of other social interactions, including related to trade and information flow, are shared within urban spaces [5,6,7,8]”.

: The sentence is a bit confusing with « including related » , is it “including those related”?

It analyzes urban street layouts and their centrality values, using a tool developed by this article and made available in the supporting materials, measuring how they scale relative to urban area estimates, that is occupation size for a given period. “

: Is it “how they scale relatively to urban area” ?

“Urban locations are assessed based on their street organization, including orthogonal (i.e., rectilinear, planned or grid-pattern streets), organic, that is self-organizing streets that generally develop around neighborhoods, and hybrid streets, which have a combination or mixture of orthogonal and organic street networks [11].”

: The end of the sentence is not necessary (it is implied), it would be better to remove it as the sentence is already really long.

“These centrality measures are compared for the different types of urban settings analyzed, with orthogonal and organic or hybrid used as the two main analytical categories due to structural similarities between hybrid and organic sites.”

: Is it for the analyzed urban settings"?

“Results for network centrality distributions, centrality, and urban area scaling”

: maybe make clearer what the second term of “centrality” is referring to (characterization of the structure)

: if needed, this is a suggestion of article about the the characterization of road network, with a method derived from spatial syntax ensuring the independence from border effect : Lagesse, C., Bordin, P., & Douady, S. (2015). A spatial multi-scale object to analyze road networks. Network Science, 3(1), 156-181.

Background

“Patterns of how networks of space are used are also analyzed, such as land use, transport, or security”

: typo (are used are also analyzed)

“In the analysis of street networks, a variety of techniques to understand urban and non-urban roads are used, with graph analysis being the most common set of methods applied for understanding street network relationships and relationships of regions and sub-regions in defined settings [18, 19].”

: repetition (“relationships and relationships”)

Scaling Approach

“This means that, although we would expect the geometry of the street network to be related to (i.e., a product of and a constraint on)city area,there is no reason for believing that any of the network values that are discussed below should take on a specific baseline value or that this baseline value will change significantly over time. This suggests, in turn, that it is also legitimate to collapse examples from different historical and geographical periods into a single scaling relationship.”

: long and unclear

: typos (spaces missing)

Results

Centrality Distributions

“Additionally, the distributions are compared using a Mann-Whitney-Wilcoxen test with a Holm–Bonferroni method [59], where all distributions, within and between the categories, showed significant

differences at p-value<0.01 levels.”

: typo (Mann-Whitney-Wilcoxon)

: I suggest to be careful with those kind of tests because they are quite sensitive to the scale of the sample. I was more expecting a correlation matrix to insure the independence of variables...

“For Figures 4 and 5, we note that the x- and y-axes are different in the figures due to distribution variations that made the same ranges difficult to apply.”

: you should make clearer that the focus is on the shape here, independently from the scale

7. PLOS authors have the option to publish the peer review history of their article (what does this mean?). If published, this will include your full peer review and any attached files.

Reviewer #1: **Yes: **Horacio Samaniego

Reviewer #2: **Yes: **Claire Lagesse

---

## [Editor Report · Acceptance letter]

29 Oct 2021

PONE-D-21-15963R1 

The structure, centrality, and scale of urban street networks: Cases from Pre-Industrial Afro-Eurasia 

Dear Dr. Altaweel:

I'm pleased to inform you that your manuscript has been deemed suitable for publication in PLOS ONE. Congratulations! Your manuscript is now with our production department. 

Kind regards, 

on behalf of

Dr. Riccardo Gallotti 

Academic Editor

PLOS ONE